

# clrDV: a differential variability test for RNA-Seq data based on the skew-normal distribution

Hongxiang Li[1] and Tsung Fei Khang[1,2]

[1] Institute of Mathematical Sciences, Universiti Malaya, Kuala Lumpur, Malaysia
[2] Universiti Malaya Centre for Data Analytics, Universiti Malaya, Kuala Lumpur, Malaysia

## ABSTRACT

**Background**. Pathological conditions may result in certain genes having expression variance that differs markedly from that of the control. Finding such genes from gene expression data can provide invaluable candidates for therapeutic intervention. Under the dominant paradigm for modeling RNA-Seq gene counts using the negative binomial model, tests of differential variability are challenging to develop, owing to dependence of the variance on the mean.

**Methods**. Here, we describe clrDV, a statistical method for detecting genes that show differential variability between two populations. We present the skew-normal distribution for modeling gene-wise null distribution of centered log-ratio transformation of compositional RNA-seq data.

**Results**. Simulation results show that clrDV has false discovery rate and probability of Type II error that are on par with or superior to existing methodologies. In addition, its run time is faster than its closest competitors, and remains relatively constant for increasing sample size per group. Analysis of a large neurodegenerative disease RNA-Seq dataset using clrDV successfully recovers multiple gene candidates that have been reported to be associated with Alzheimer's disease.

# INTRODUCTION

## Background

Finding patterns of gene expression variation that are associated with a biological condition of interest is the first step towards elucidating the molecular basis underlying a biological process. Currently, bulk tissue mRNA collected under specific biological conditions through RNA-sequencing (RNA-Seq) technologies remains an important approach for studying gene expression patterns. Typically, genes that show statistically and biologically meaningful difference in mean expression between conditions are of interest. Indeed, pathological conditions frequently manifest as gene sets with altered mean mRNA expression levels. The identification of these genes is important for understanding how the functions of normal molecular pathways are perturbed (*Van den Berge et al., 2019*). Hence, detecting genes that are differentially expressed is a routine and main use of RNA-Seq data

Corresponding author
Tsung Fei Khang,
tfkhang@um.edu.my

(*Stark, Grzelak & Hadfield, 2019*). To analyse differential gene expression, a multitude of statistical tests have been developed throughout the years. Methods such as edgeR (*Robinson, McCarthy & Smyth, 2010*), DESeq2 (*Love, Huber & Anders, 2014*) and voom (*Law et al., 2014*) have become established, go-to methods for differential expression (DE) analysis.

To obtain a more complete picture of patterns of gene expression variation, we need to look beyond genes with significantly different mean expression (DE genes) between conditions (*Gorlov et al., 2012*). We refer to the class of genes that show biologically meaningful differences in expression variability (EV) as differential variability (DV) genes. Genes can influence each other's expression through regulatory interactions, and noise in the expression of one gene can propagate to downstream genes (*Raser & O'Shea, 2005*). Consequently, gene EV plays a crucial role in the organization of regulatory circuits and signal transduction pathways (*Mar et al., 2011*; *Komurov & Ram, 2010*). Furthermore, the position of proteins within the signaling network hierarchy strongly correlates with their EV and functional centrality, reflecting a generic mechanism of transcriptional regulation of the cellular signaling network (*Komurov & Ram, 2010*). *Mar et al. (2011)* observed that genes with lower EV tend to be core members of pathways and more connected to other network members, while genes with higher EV have fewer connections. Specifically, in disease states such as Parkinson's disease, increased EV in core signaling pathways can lead to dysregulation and diminish the network's robustness to external events. Therefore, identifying DV genes can shed light on the underlying mechanisms of complex biological phenomena. For example, in neurobiology, genes that show differential variability between undifferentiated and differentiating states have been found to be associated with body axis development, neuronal movement, and transcriptional regulation during the neural differentiation process (*Ando, Kato & Honda, 2015*). In cancer biology, DV genes are useful as biomarkers for predicting tumor progression and prognosis (*Dinalankara & Corrada Bravo, 2015*), and patient survival (*Strbenac et al., 2016*). *Gorlov et al. (2012)* found that genes with larger expression variance in tumors compared to normal cells show stronger association with clinically important features. Finally, increased gene expression variability is a common outcome of aging (*Bahar et al., 2006*; *Stegeman & Weake, 2017*). Standard DE analyses are likely to miss DV gene candidates, since they are not optimized for detecting differences in expression variability.

The general idea of conducting a test of differential variability for RNA-Seq data involves testing the equality of variances (equivalently, standard deviations) between two populations. The variance parameter is embedded in some probability distribution that approximates the distribution of gene (more generally, transcript) counts, assuming the null hypothesis is correct. The standard approach models RNA-Seq data as a discrete random variable.

Before modeling can be done, the raw count data need to be normalized to account for variation in the sequencing depth of each sample. Commonly used methods include the trimmed mean of M values (TMM) (*Robinson & Oshlack, 2010*), the median-of-ratios method (*Anders & Huber, 2010*; *Love, Huber & Anders, 2014*), upper-quartile (*Bullard et al., 2010*), conditional quartile normalization (*Hansen, Irizarry & Wu, 2012*), *etc.* After this, a model that accounts for overdispersion commonly seen in RNA-Seq data (*e.g.*, the

negative binomial distribution) is used, but alternative models are possible (*Esnaola et al., 2013*). Statistical tests of differential variability can then be based on estimators of suitable model parameters for representing expression variability.

To date, only a few methods are available for finding DV genes using RNA-Seq data. In contrast, even in 2015, there were 22 methods for detecting DE genes (*Khang & Lau, 2015*). For testing differential variability of genes between two populations using RNA-Seq data, initial methods co-opted techniques from microarray data analysis. DiffVar (*Phipson & Oshlack, 2014*) is an empirical Bayes method that depends on the limma (*Smyth, 2005*) framework. Subsequently, the negative binomial (NB) model with two different parametrizations (NB1, NB2; see *Cameron & Trivedi (2013)*) from DE tests was used for developing DV tests. MDSeq (*Ran & Daye, 2017*) uses the coefficient of dispersion ($\sigma^2/\mu$) from a generalized linear model with NB1 parametrization as a measure for variability. Accordingly, the variance is represented as the product of the mean $\mu$ and the dispersion $\phi$ parameter, $\sigma^2 = \phi\mu$. The parameter $\mu$ is viewed as a technical component, whereas $\phi$ is treated as a biological component and interpreted as a parameter for gene expression variability. *De Jong, Moshkin & Guryev (2019)* proposed a DV test that uses the generalized additive models for location, scale and shape (GAMLSS; *Rigby & Stasinopoulos, 2005*) framework for quantifying expression variability. GAMLSS uses the NB2 parametrization, whereby the variance and the mean are related quadratically as $\sigma^2 = \mu + \phi\mu^2$. Recently, *Roberts, Catchpoole & Kennedy (2022)* developed DiffDist, a hierarchical Bayesian model based on the NB2 model. In their work, gene expression variability is measured using the dispersion parameter $\phi$, which is treated as a log-normal prior. Subsequently, test of difference in dispersion between two conditions is based on the posterior distribution simulated using Markov Chain Monte Carlo (MCMC).

In recent years, there has been an increasing call towards adopting a compositional data analysis (CoDA) framework for improving the analysis of RNA-Seq data. Compositional data analysis originated from the study of chemical, mineral, and fossil compositions of rocks and sediments (*Aitchison, 1981*). In this type of analysis, observations are expressed as proportions relative to a total, with each part referred to as a component that sums to unity. CoDA analyzes the relative differences between the components instead of their absolute values. In the closely related field of microbiome data analysis, CoDA forms the main theoretical framework of data analysis and differential abundance methods (*Gloor et al., 2017*). Nevertheless, the diffusion of CoDA approach into RNA-Seq data analysis is slow, possibly because established protocols for routine analyses such as differential expression analysis (*e.g.*, DESEq2, edgeR) are all based on discrete count models such as the NB model. *Quinn et al. (2018)* argued that next-generation sequencing abundance data should be viewed as inherently compositional because only a portion of genes may be sampled by sequencers, and cells are likely to be constrained in their capacity for mRNA production. Furthermore, *Quinn, Crowley & Richardson (2018)* showed the feasibility of applying ALDEx2 (*Fernandes et al., 2014*), a tool developed for differential abundance analysis in microiobiome studies under a CoDA framework, to differential expression analysis using RNA-Seq data. Encouragingly, they reported that ALDEx2 shows superior performance with respect to precision and recall when compared against edgeR and

DESeq2. By removing the need to rely on assumptions that justify normalization protocols in standard count-based approaches, log-ratio based transformations of RNA-Seq data in compositional form is potentially more attractive and effective for differential expression analyses (*Quinn et al., 2019*). More recently, *McGee et al. (2019)* developed absSimSeq - a novel simulation protocol for generating realistic RNA- Seq data using a compositional data framework.

In this article, we propose clrDV (centered log-ratio transformation-based test for Differential Variability), a novel method for detecting DV genes between two conditions in RNA-Seq data that is based on a compositional data analysis framework. The method involves a log-ratio transformation (*Aitchison, 1986*) of the raw gene counts, which results in a continuous variable. We model the distribution of the transformed variable using the skew-normal distribution (*Azzalini, 1985*) with centered parameters. Subsequently, we construct a Wald test statistic for testing differential variability. Through simulations, we show how well clrDV performs compared to existing methods. Finally, we demonstrate the applied value of clrDV by using it to identify biologically meaningful genes in the analysis of a large RNA-seq dataset from a neurodegenerative disease study.

## MATERIALS AND METHODS

### The centered log-ratio transformation

We begin with the assumption that RNA-seq data are compositional, *i.e.,* each sample is represented by a vector of relative frequencies of the genes, which sums to unity. The key step in processing compositional data involves a log-ratio transformation, for which several variants are available. The simplest is the centered log-ratio (CLR) transformation, first proposed by *Aitchison (1986)*. After CLR-transformation, the simplex space of the compositional data is transformed into the Euclidean space. It is then convenient to view CLR-transformed values as realizations of a continuous random variable. To be concrete, let $X_{gi}$ be the read count for gene $g$ and sample $i$, where $g = 1, 2, \ldots, G$ and $i = 1, 2, \ldots, n$. For a $G$-component composition $\{x_{1i}, x_{2i}, \ldots, x_{Gi}\}$, the CLR- transformation of $X_{gi}$ is given by

$$\mathrm{CLR}(X_{gi}) = \log\left\{\frac{x_{gi}}{(\prod_{g'} x_{g'i})^{1/G}}\right\} = \log(x_{gi}) - \frac{1}{G}\sum_{g'=1}^{G}\log(x_{g'i}),$$

for $g' = 1, 2, \ldots, G$. We call $\mathrm{CLR}(X_{gi})$ the relative gene expression, or CLR-transformed count, of gene $g$ and sample $i$. A pseudo-value 0.5 is added if $x_{gi} = 0$ for any $i$. Thus, the main challenge for using CLR-transformed data to develop a test for differential variability is modeling them using a tractable probability distribution for which estimation of the variance parameter is practical.

### The skew-normal model for modeling centered log-ratio transformed data

The skew-normal distribution is a three-parameter continuous probability distribution that generalizes the normal distribution. Its additional skewness parameter enables it

to model skewed data on the real line, thus making it more flexible than the normal distribution, while retaining the usual mean and variance parameters. Historically, the skew-normal model was arrived at by several different authors in other contexts (*e.g.*, as a prior distribution in Bayesian analysis by *O'Hagan & Leonard (1976)*; see *Azzalini (2022)*). Its main theoretical properties were developed by *Azzalini (1985)*.

Denote the relative gene expression from gene $g$ in sample $i$ by $Y_{gi}$. We model $Y_{gi}$ using a skew-normal distribution with centered parameters (CP), that is, $Y_{gi} \sim \mathrm{SN_C}(\mu_g, \sigma_g, \gamma_g)$, where $\mu_g$ is the mean, $\sigma_g$ is the standard deviation, and $\gamma_g$ is the skewness parameter, $g = 1, 2, \ldots, G$ and $i = 1, 2, \ldots, n$. The parameter vector $\boldsymbol{\theta}g^{(C)} = (\mu_g, \sigma_g, \gamma_g)$ has parameter space $\mathbb{R} \times \mathbb{R}^+ \times (-k, k)$, where $k = \sqrt{2}(4-\pi)/(\pi-2)^{3/2} \approx 0.9953$. The special case of $\gamma_g = 0$ results in a normal distribution with mean $\mu_g$ and variance $\sigma_g^2$. See *Li & Khang (2022)* for initial description and Supplementary Material S1 for further mathematical details.

To perform parameter estimation and carry out related numerical tasks involving the skew-normal distribution, we used the sn (*Azzalini, 2022*) R package. Regular maximum likelihood estimation of parameters of the skew-normal model was first done using the function selm(). If NA values were returned, we used the maximum penalized likelihood estimation as implemented using the Qpenalty option. If NA values persisted, the MPpenalty option was used.

For RNA-Seq experiments comparing two populations, testing for differential variability is equivalent to testing the equality of the standard deviation of relative gene expressions in two populations, that is, $\sigma_{g,1} = \sigma_{g,2}$. For this purpose, we can use the Wald statistic

$$Z_g = \frac{\hat{\sigma}_{g,2} - \hat{\sigma}_{g,1}}{\sqrt{\mathrm{Var}(\hat{\sigma}_{g,2}) + \mathrm{Var}(\hat{\sigma}_{g,1})}},$$

for $g = 1, 2, \ldots, G$, where $\hat{\sigma}_{g,j}, j = 1, 2$ are the maximum likelihood estimators of the standard deviation of the skew-normal distribution with centered parameters for population 1 and population 2, and $\mathrm{Var}(\hat{\sigma}_{g,j}), j = 1, 2$ are the corresponding diagonal elements of the estimated Fisher information matrix of centered parameters $\boldsymbol{\theta}g^{(C)}$. The Wald statistic converges in distribution to the standard normal distribution as sample size becomes large.

To control the false discovery rate (FDR) as a result of conducting multiple hypothesis tests across genes, we applied the Benjamini-Yekutieli procedure (*Benjamini & Yekutieli, 2001*), which allows for arbitrary dependence between the tested hypotheses. Note that in the context of samples, FDR is estimated as the sample proportion of false discoveries.

## Data description and preprocessing

In order to evaluate the suitability of the skew-normal distribution for fitting the CLR-transformed counts and compare the performance of clrDV and other methods in terms of controlling false discovery rate (FDR) and probability of Type II error, it is important to simulate the null distribution using realistic parameter values. For this purpose, we used two real RNA-Seq datasets that have modest to large sample sizes to enable reliable estimation of the parameters of the NB2 model. The first dataset (GEO accesion number: GSE123658) contains whole blood RNA-Seq data from from 39 Type 1 diabetes patients

and 43 healthy donors (*Leal Valentim et al., 2020*), with 16,785 transcripts. The second dataset (GEO acccesion number: GSE150318) contains longitudinal gene expression data from 114 short-lived killfish *Nothobranchius furzeri* measured at 10 weeks and 20 weeks of age (*Kelmer Sacramento et al., 2020*), with 26,739 transcripts. Hereafter, we call these two datasets the "Valentim dataset" and the "Kelmer dataset".

For empirical assessment, we used the Mayo RNASeq dataset (*Allen et al., 2016*), which consists of 278 samples and 64,253 transcripts. In this study, RNA was isolated from the temporal cortex of brains of patients with four biological conditions: control ($n = 80$), Alzheimer's disease (AD; $n = 84$), progressive supranuclear palsy (PSP; $n = 84$) and pathologic aging ($n = 30$). We chose to compare the control group against the AD and the PSP group respectively, since the sample sizes in these groups are reasonably large and balanced. The large sample sizes per group of this dataset enables reliable estimation of the variance parameter in the skew-normal model. In addition, since Alzheimer's disease is a well-researched neurodegeneratve disease, literature support for association with Alzheimer's disease may be more readily found for the DV genes detected, thus enhancing interpretability of the analysis results.

For gene filtering, we removed a gene if it has average count-per-million (CPM) below 0.5, or its count is zero in at least 85% of the samples. After filtering, a total of 12,283 and 16,670 transcripts were left for the Valentim dataset and the Kelmer dataset, respectively. In the Mayo RNA-Seq dataset, after removing samples with missing class labels, 78, 82, and 84 samples were left for the control, AD and PSP groups, respectively. For the AD-control comparison, a total of 18,664 transcripts were left; for the PSP-control comparison, $18,636$ transcripts were left. For MDSeq, diffVar, GAMLSS, and DiffDist, we normalized the raw counts using TMM normalization.

## Simulation study

Our simulation study has two objectives: (1) evaluation of the goodness-of-fit of the skew-normal distribution on CLR-transformed data; (2) comparison of the performance of clrDV against MDSeq, diffVar, GAMLSS (Benjamini–Hochberg (BH) and Benjamini–Yekutieli (BY) variants), and DiffDist with respect to FDR and probability of Type II error.

For objective (1), we randomly chose 10,000 of the filtered genes and modeled the distribution of RNA-Seq counts for each of them using the NB2 model. This was done by first estimating the parameters of the NB2 model using samples from one group (Type 1 diabetes group in the Valentim dataset; 20-weeks group in the Kelmer dataset), followed by simulation of 500 biological replicates from the NB2 models with estimated parameters. We used the `polyester` R package (*Frazee et al., 2015*) to perform the simulation. After gene and sample filtering using the mean CPM $< 0.5$ and proportion of zero count $\geq 0.85$ criteria, we checked the goodness-of-fit of the skew-normal model on the distribution of the CLR-transformed simulated count data using the Kolmogorov–Smirnov (KS) test.

For objective (2), we first filtered out genes and samples using the mean CPM $< 0.5$ and proportion of zero count $\geq 0.85$ criteria. We then randomly selected 2,000 of the remaining genes and estimated the parameters of the NB2 model for each gene, again using

the `polyester` R package. Thus, in a two-group comparison setting, both groups have identical NB2 model parameters. For 10% of the genes (200/2,000), we simulated them as DV genes by multiplying their estimated size parameter $(1/\phi)$ in one of the two groups with a random value $x$, where $x \in (0.25, 0.5) \cup (2, 4)$. For DiffDist, a random subset of a quarter of the DV(50/200) and non-DV genes (450/1,800) were used, since the MCMC step in DiffDist is computationally expensive. Subsequently, biological replicates of size 50, 100, 150, 200 per group were drawn from these NB2 models. The minimum sample size considered was 50 because for smaller values, estimation of the variance parameter of the skew-normal model is unreliable (*Azzalini & Capitanio, 2014*; *Azzalini, 2022*). A total of 30 instances were thus simulated, and the methods considered were used to test for DV genes. Genes with BY-adjusted $p$-value $< 0.05$ were flagged as having differential variability. We evaluated the DV tests considered by plotting the probability of Type II error against FDR for the 30 simulation instances. Finally, we recorded the run times of each method.

### Empirical assessment

We applied clrDV to the Mayo RNA-Seq dataset to assess its capacity for detecting DV genes that are contextually meaningful. For the genes after filtering, we assessed the fit of the skew-normal model to their CLR-transformed count data using the KS-test. Comparisons were made to close competitors as indicated from the results of the simulation study. Volcano plots were used to inspect the biological effect size and statistical significance of all genes tested. Venn diagrams were used to identify sets of genes that are identically recovered by all three methods, by combinations of two methods, or uniquely recovered by a single method. Violin plots of selected DV genes were made to verify computational results.

### Tools and computing environment

Computational tasks were done in a computer with a 1.80 GHz i5-8265U CPU and an 8GB RAM processor. R version 4.2.1 (*R Core Team, 2022*) operating in Windows 10 was used. The complete list of R packages used is given in Supplemental Information 2. ENSEMBL gene ID to gene symbol conversion was done using the application programming interface of the BioTools.fr website (*Saurin, 2022*).

## RESULTS

### Simulation study

For the skew-normal model to adequately capture variation in the CLR-transformed RNA-Seq data, it is necessary to demonstrate goodness-of-fit between model and observed data. Figure 1 shows the distribution of the $p$-value of KS tests for the fit of the skew-normal model on distribution of the CLR-transformed count data. For the majority of the genes (96.4% and 97.3% genes in the simulated Valentim and the Kelmer datasets, respectively), the fit is good ($p$-value $> 0.05$).

Next, to be a viable alternative to existing methods for detecting differential variability, clrDV should have comparable or superior performance with respect to FDR and the probability of Type II error across sample size scenarios appropriate for the testing of

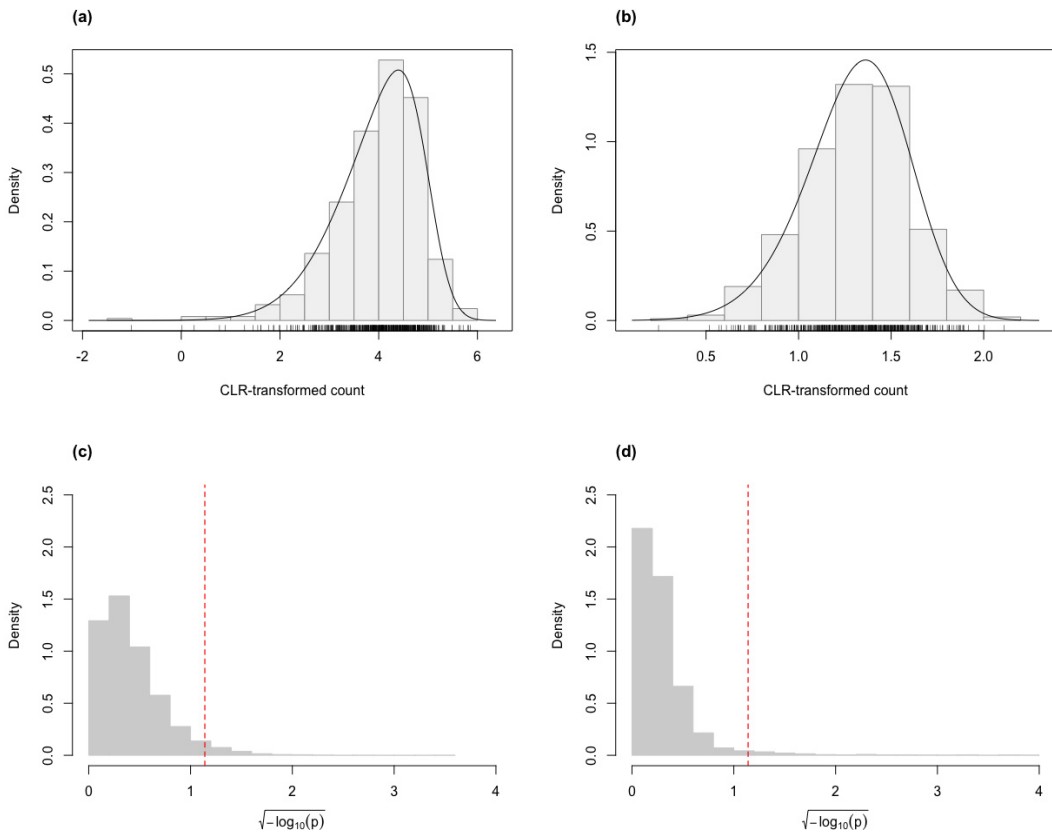

**Figure 1** Histograms of CLR-transformed counts for two genes with fitted skew-normal curve for (A) the Valentim dataset ($\hat{\mu} = 3.968$ (s.e. = 0.038), $\hat{\sigma} = 0.858$, (s.e. = 0.030) and $\hat{\gamma} = -0.732$ (s.e. = 0.055); (B) the Kelmer dataset ($\hat{\mu} = 1.311$ (s.e. = 0.013), $\hat{\sigma} = 0.281$ (s.e. = 0.009) and $\hat{\gamma} = -0.331$ (s.e. = 0.107)). Distribution of the *p*-values (after $\sqrt{-\log_{10} p}$ transformation for compactness) of the Kolmogorov–Smirnov goodness-of-fit tests of the skew-normal model for genes in the simulated (C) Valentim dataset and (D) Kelmer dataset. The skew-normal model gives good fit to about 96.4% and 97.3% of the genes in (C) and (D), respectively. The red dashed line corresponds to the threshold *p*-value of 0.05.

differential variability. Figure 2 shows the scatter plots of probability of Type II error against FDR for analysis of the simulated Valentim dataset, for each of the four sample sizes (50, 100, 150, 200) per group scenarios. For sample size of 50, all methods except GAMLSS-BH show relatively larger mean probability of Type II error ($>0.5$), indicating low statistical power to detect DV genes. GAMLSS-BH shows an acceptable mean probability of Type II error of about 0.2, with a trade-off in increased mean FDR of about 0.05. diffVar shows highly inflated FDR. For other sample sizes, clrDV is uniformly superior against MDSeq with respect to mean FDR and mean probability of Type II error; against GAMLSS-BH, clrDV has uniformly superior mean FDR; against GAMLSS-BY, clrDV gives approximately similar mean FDR and mean probability of Type II error. With respect to computing speed, clrDV is substantially faster than GAMLSS (both BH and BY variants) as sample size increases (Table S1). Overall, diffVar is uniformly inferior to all other methods (mean probability of Type II error $> 0.05$ and FDR $> 0.17$, for all sample sizes). DiffDist

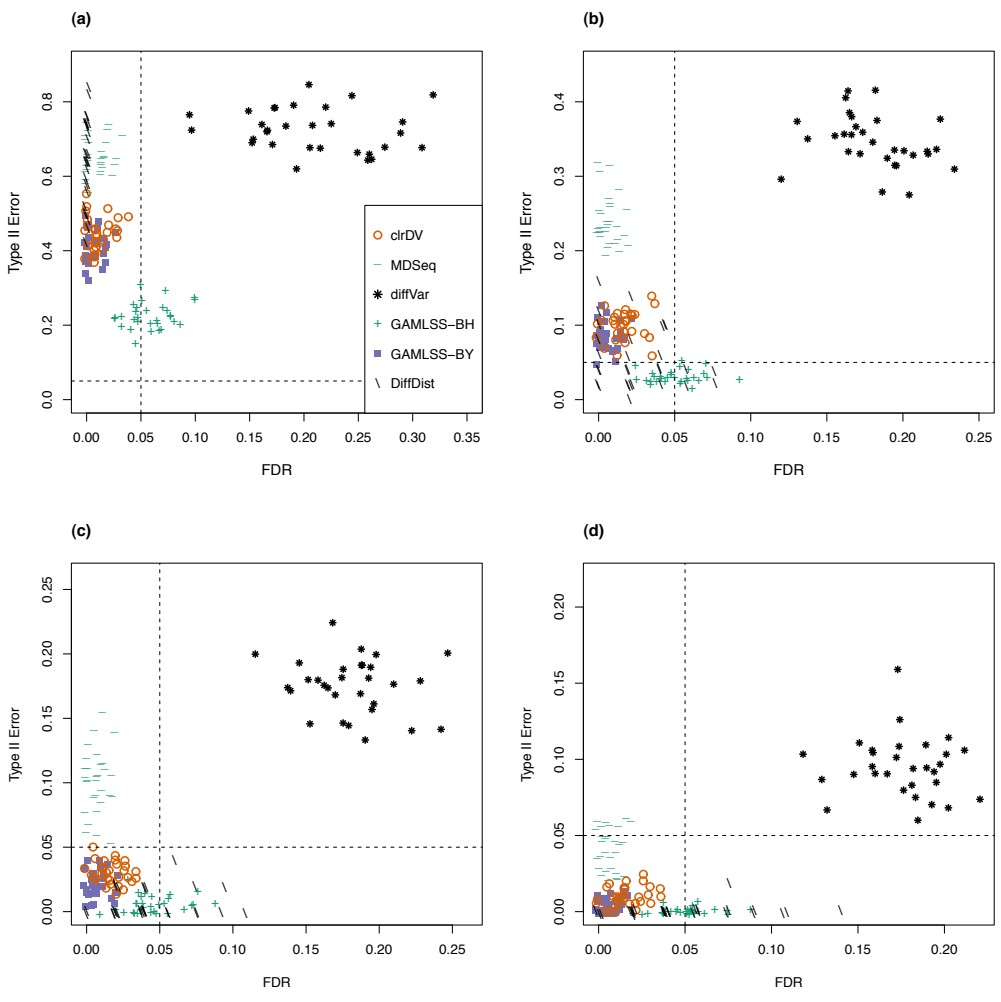

**Figure 2** **Scatter plots of probability of Type II error *vs.* FDR for simulation study of the Valentim dataset (30 instances) for samples size per group of (A) 50, (B) 100, (C) 150, and (D) 200.** Dashed lines represent probability of Type II error and FDR of 0.05.

appears to be more aggressive in calling DV genes, at the cost of having large variation in for FDR, particularly for sample sizes of 150 or more. It also has the longest run time among all methods.

For the analysis of simulated data from the Kelmer dataset, we find clrDV to have comparable mean FDR and mean probability of Type II error (Fig. 3) as MDSeq and GAMLSS-BY. However, clrDV computing time remains almost constant across the four sample sizes (50, 100, 150, 200), whereas MDSeq, GAMLSS and DiffDist have computing times that increase with sample size (Table S2). diffVar, GAMLSS-BH and DiffDist are inferior in controlling FDR across all four sample sizes.
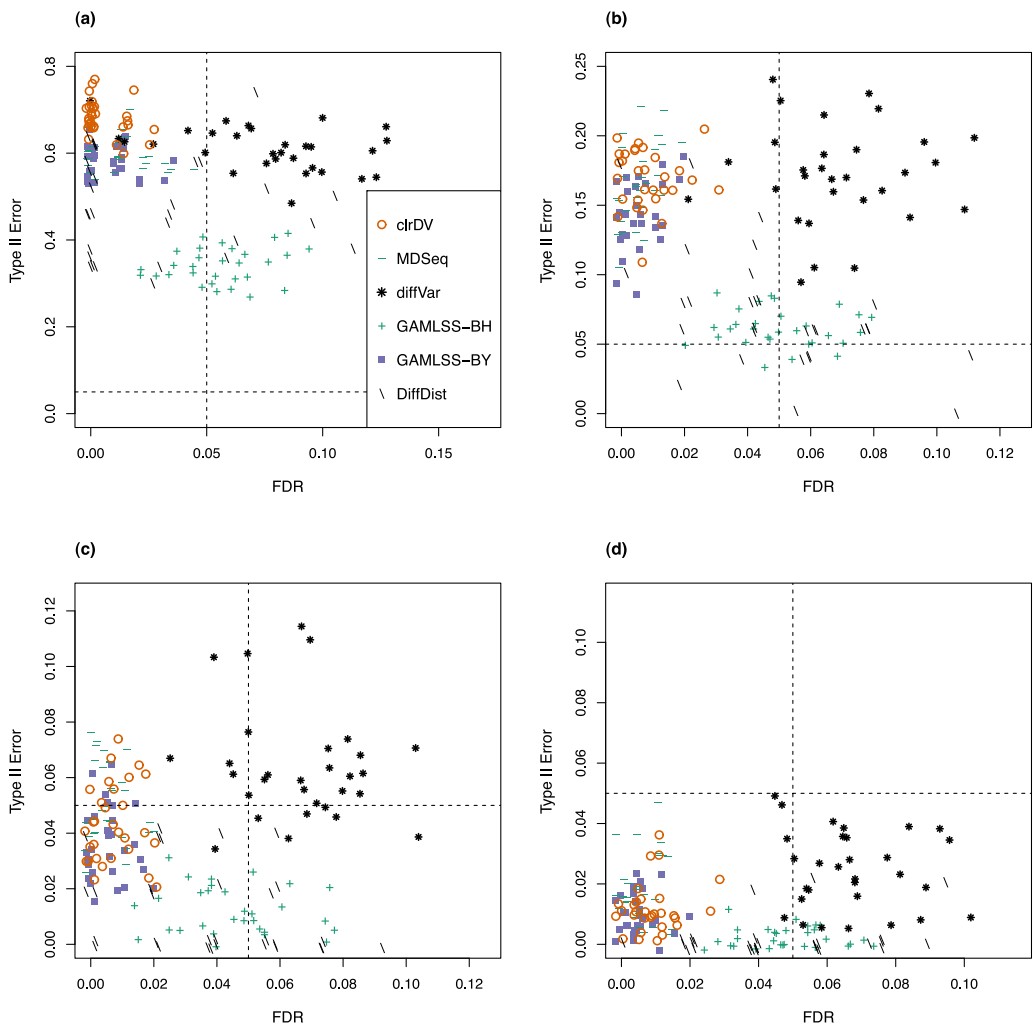

**Figure 3** **Scatter plots of probability of Type II error *vs.* FDR for simulation study of the Kelmer dataset (30 instances) for samples size per group of (A) 50, (B) 100, (C) 150, and (D) 200.** Dashed lines represent probability of Type II error and FDR of 0.05.

## Detection of genes with differential variability

Similar to our findings from simulated data, the skew-normal model also appropriately captures variation in the CLR-transformed counts of the genes for the Mayo RNA-Seq dataset (Fig. 4). Almost all genes have KS goodness-of-fit test *p*-value greater than 0.05 (98.5% and 99.5% for the control and the AD groups, respectively; 98.4% and 99.2% for the control and the PSP groups, respectively.)

Applying the procedure described in Section 'The skew-normal model for modeling centered log-ratio transformed data', we estimated the standard deviation of the CLR-transformed data, computed the Wald statistic and subsequently the BY-adjusted *p*-value for each tested gene. For the control *vs.* AD comparison, we detected a set of 4,754 DV genes (see Table S3 for complete list); for the control *vs.* PSP comparison, 4,859 DV

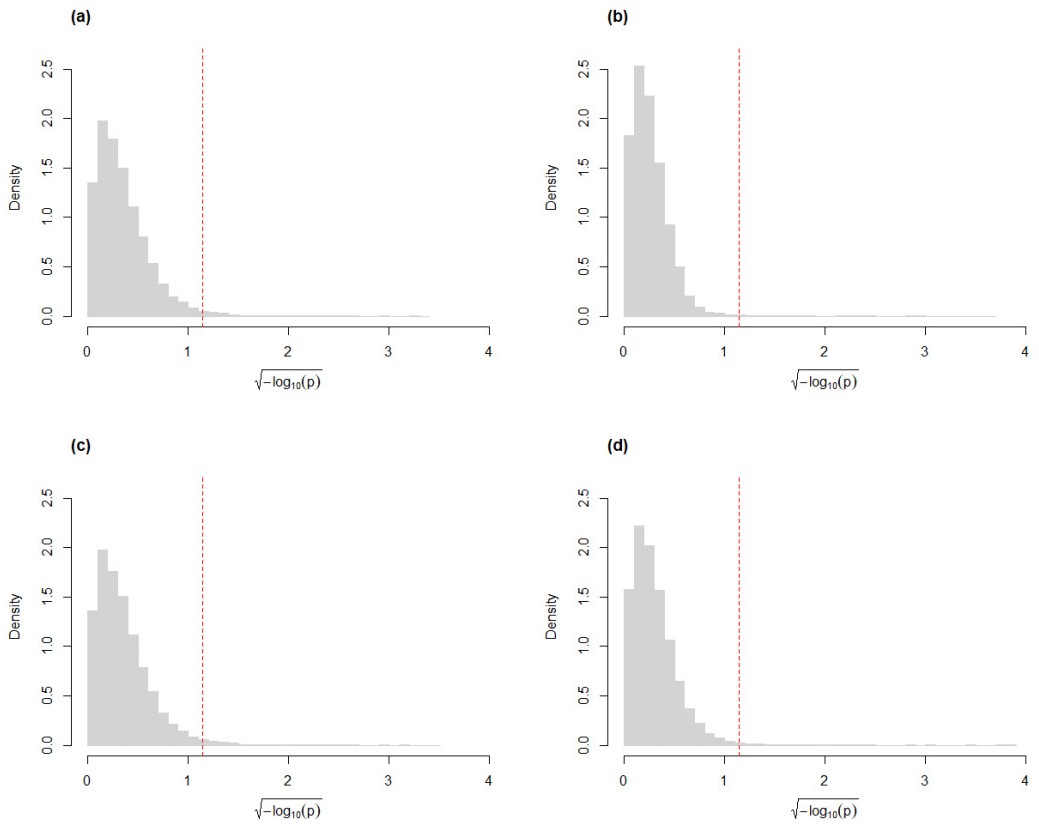

**Figure 4** Histograms of the distribution of the *p*-values (after $\sqrt{-\log_{10}p}$ transformation for compactness) from Kolmogorov–Smirnov goodness-of-fit tests of the skew-normal model on genes from the (A) the control group (*p*-value > 0.05 for 98.5% of genes) and (B) the AD group (*p*-value > 0.05 for 99.5% of the genes) of the control *vs.* AD comparison; (C) the control group (*p*-value > 0.05 for 98.4% of genes) and (D) the PSP group (*p*-value > 0.05 for 99.2% of genes) of the control *vs.* PSP comparison. The red dashed line corresponds to the threshold *p*-value of 0.05.

genes were detected (see Table S4 for complete list). For the majority of DV genes, the estimated standard deviation in the control group is larger than the one in the treatment group (Fig. 5). This observation suggests that there is a higher proportion of DV genes with reduced expression variability among AD patients compared to genes with increased variability. In fact, many genes associated with AD may have a limited range of expression levels, often being abnormally high or low, which could play a role in maintaining the pathological state of the disease.

Only MDSeq and GAMLSS (BH and and BY variants) were considered for empirical assessment since the results of the simulation study indicate that they are the only reasonable competitors of clrDV with respect to appropriate control of FDR control and probability of Type II error at sample sizes close to 100.

Figure 6 shows the number of significant DV genes identified by clrDV, MDSeq, GAMLSS-BH and GAMLSS- BY for the control *vs.* AD comparison (see Table S5 for complete list). GAMLSS-BH detected the most DV genes (9,926), followed by MDSeq (6,924), and clrDV (4,754). The high confidence gene set, defined as the intersection
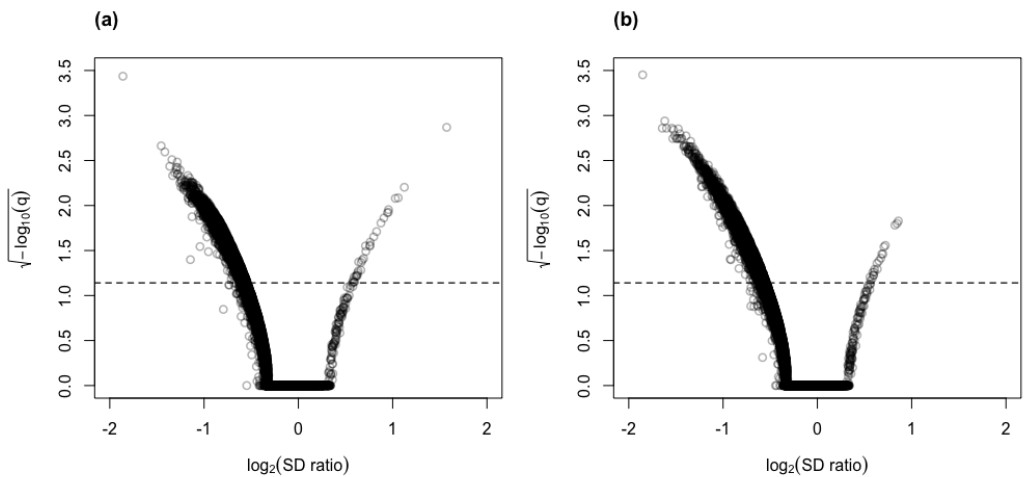

**Figure 5** **Volcano plots for (A) control *vs.* AD and (B) control *vs.* PSP comparisons for the Mayo RNA-Seq dataset.** Dashed line represents the threshold of BY- adjusted *p*-value ($q$) at 0.05 for flagging DV genes. The number of DV genes with $\log_2(\text{SD ratio}) > 0$ and $\log_2(\text{SD ratio}) < 0$ respectively: (A) 32 and 4,722; (B) 19 and 4,840.

of DV genes from each method, contains genes with estimated $\log_2(\text{SD ratio})$ that is relatively large ($>0.5$). About 99.8% (4,743/4,754) of DV genes detected by clrDV are also identified by MDSeq or GAMLSS-BH; 92.0% (4,374/4,754) are detected by both MDSeq and GAMLSS-BH; about 0.2% (11/4,754) are uniquely identified by clrDV. GAMLSS-BH identified very large numbers of DV genes in this dataset, but the majority of these are probably false positives, given its relatively poorer control of FDR as shown in the results of the simulation studies. Moreover, these DV genes have estimated $\log_2(\text{SD ratio})$ with small magnitude, suggesting lack of bioloigcal significance (Fig. 6C).

Using GAMLSS-BY, only 6,079 DV genes were detected, compared to 9,926 using GAMLSS-BH. Thus, GAMLSS-BY primarily helps improve FDR by reducing the number of DV genes called. Between 61.7% (4,271/6,924) and 89.8% (4,271/4,754) of the DV genes detected by one method are detected by all three. About 97.0% (4,613/4,754) of DV genes detected by clrDV are identified by one of other two methods, and 3.0% (141/4,754) of DV genes detected by clrDV are unique.

The result of the control *vs.* PSP comparison is similar (Fig. 7; Table S6). GAMLSS-BH also detected the most number of DV genes (9,707), followed by MDSeq (6,894), and clrDV (4,859). Up to 99.4% (4,831/4,859) of DV genes identified by clrDV are detected by MDSeq or GAMLSS-BH; about 89.1% (4,329/4,859) are detected by both MDSeq and GAMLSS-BH; about 0.6% (28/4,859) are uniquely identified by clrDV. Using GAMLSS-BY, only 6,024 DV genes were flagged. Approximately 95.9% (4,658/4,859) of DV genes identified by clrDV are also identified by MDSeq or GAMLSS-BY; about 86.1% (4,186/4,859) are detected by both MDSeq and GAMLSS-BY; about 4.1% (201/4,859) are uniquely detected by clrDV.
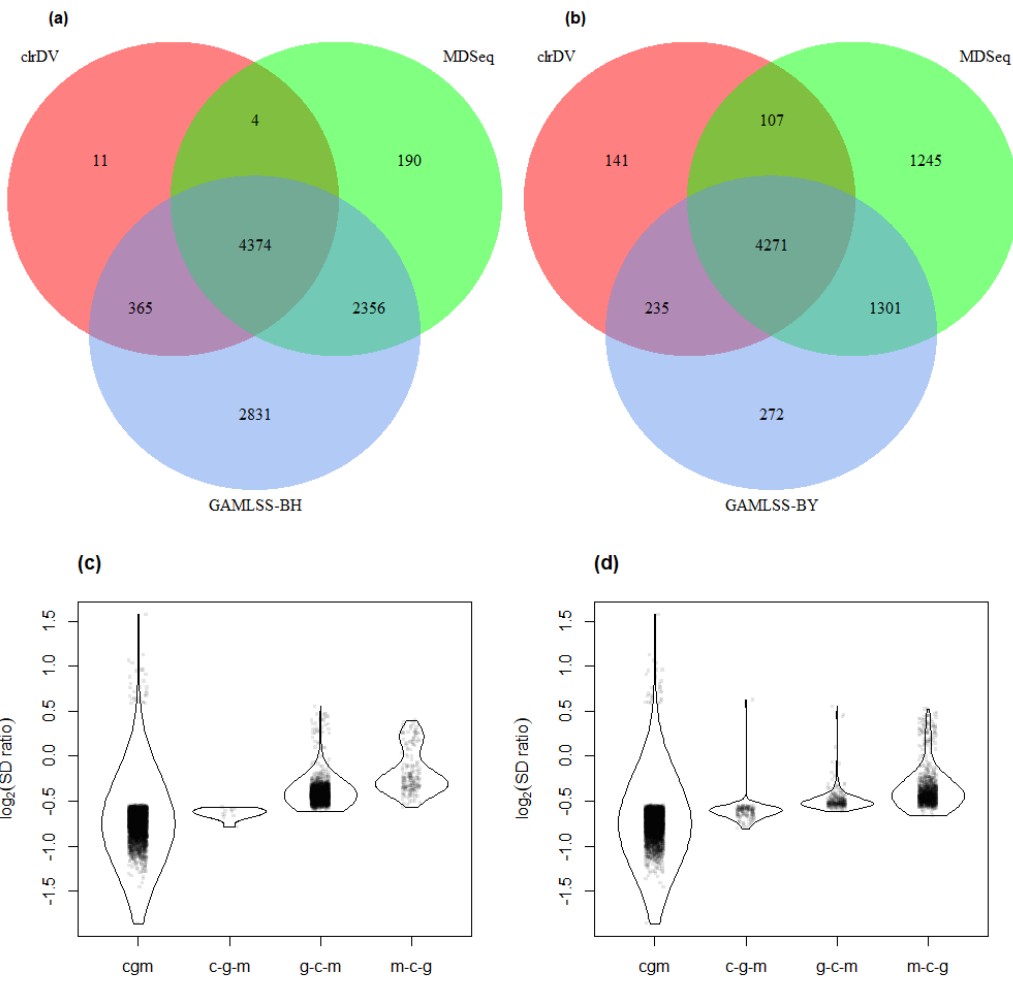

**Figure 6** **Venn diagrams of DV genes detected by clrDV, MDSeq and (A) GAMLSS-BH; (B) GAMLSS-BY for the control *vs.* AD comparison. Violin plots of the distribution of estimated log$_2$(SD ratio) of the DV genes detected using clrDV, MDSeq and (C) GAMLSS-BH; (D) GAMLSS-BY.** Abbreviations: cgm, DV genes detected by clrDV, GAMLSS and MDSeq; c-g-m, DV genes detected by clrDV only; g-c-m, DV genes detected by GAMLSS-BH only; m-c-g, DV genes detected by MDSeq only.

The violin plots (Figs. 6 and 7) suggest that the DV genes uniquely called by clrDV may be more likely to true positives, given that the magnitude of log$_2$(SD ratio), which is associated with biological significance, is generally larger than 0.5. For those genes uniquely called by GAMLSS or MDSeq, the order of magnitude is generally below 0.5. With respect to run time, for the control *vs.* AD comparison, clrDV took about 7.5 min, compared to 6 min for MDSeq, and 13 min for GAMLSS; for the control *vs.* PSP comparison, clrDV took about 7 min, while MDSeq used 6 min, and GAMLSS used 15 min.

### Biological significance of detected differential variability genes

For clrDV to be of practical use, it is necessary to show that it can successfully recover genes with reported associations with a specific biological condition. Thus, in the analysis of the Mayo RNA-Seq dataset for Alzheimer's disease, four of the DV genes detected in the
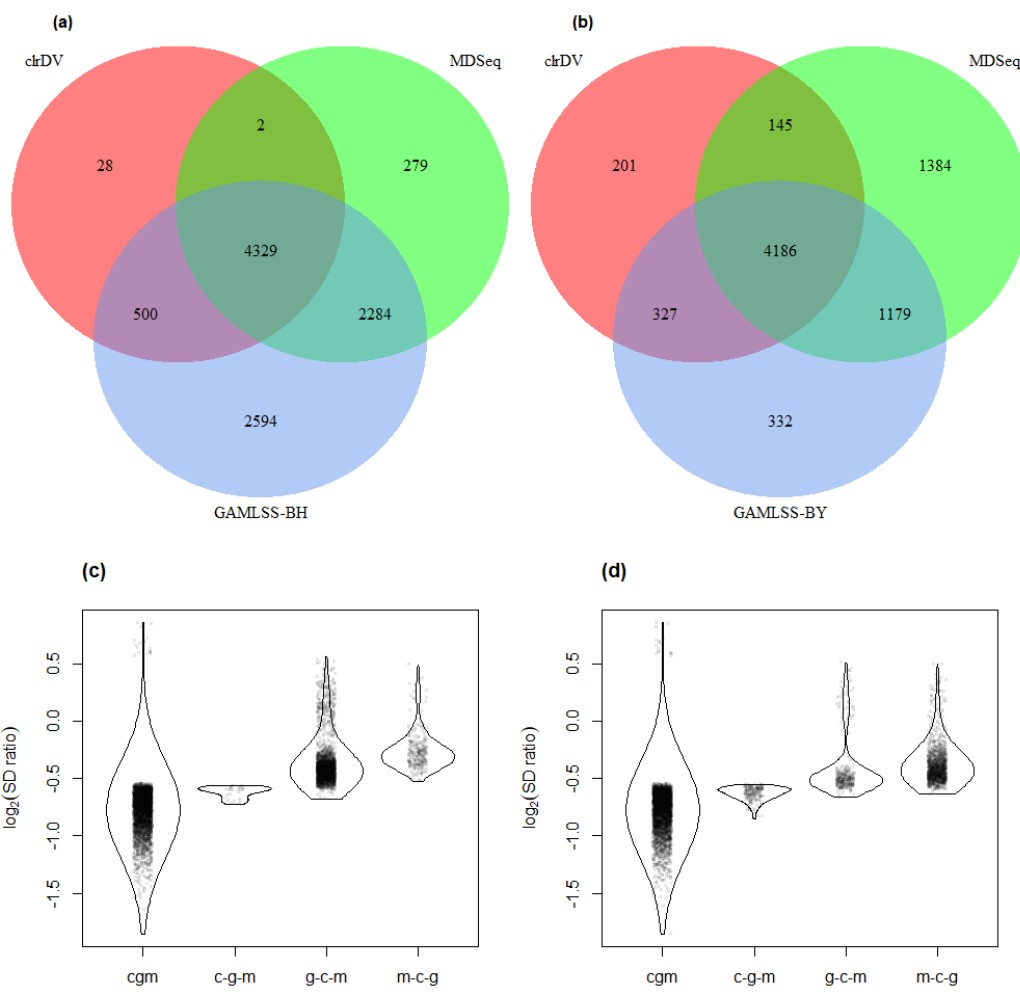

**Figure 7** **Venn diagrams of DV genes detected by clrDV, MDSeq and (A) GAMLSS-BH; (B) GAMLSS-BY for the control *vs.* PSP comparison. Violin plots of the distribution of estimated log$_2$(SD ratio) of the DV genes detected using clrDV, MDSeq and (C) GAMLSS-BH; (D) GAMLSS-BY.** Abbreviations: cgm, DV genes detected by clrDV, GAMLSS and MDSeq; c-g-m, DV genes detected by clrDV only; g-c-m, DV genes detected by GAMLSS-BH only; m-c-g, DV genes detected by MDSeq only.

control *vs.* AD comparison that have the largest estimated SD ratio above 1 are LTBP2, SLPI, C2orf40, and SLC47A1 (Fig. 8). All four genes have been reported to be associated with Alzheimer's disease in the literature. The latent transforming growth factor (TGF)- beta binding proteins (LTBP) are important components of the extracellular matrix (*Robertson et al., 2015*). They interact with fibrillin microfibrils, and are known to be mediators of TGF-$\beta$ functions (*Rifkin, Rifkin & Zilberberg, 2018*), dysfunctions of which have been implicated in Alzheimer's disease (*Das & Golde, 2006*). Then, the secretory leukocyte protease inhibitor protein(SLPI) is known to regulate the penetrance of frontotemporal lobar degeneration (FTLD) in patients who have mutations in the progranulin gene (*Ghidoni et al., 2014*). Loss of progranulin function has been found to enhance microglial neuroinflammation, which is implicated in Alzheimer's disease (*Mendsaikhan, Tooyama*

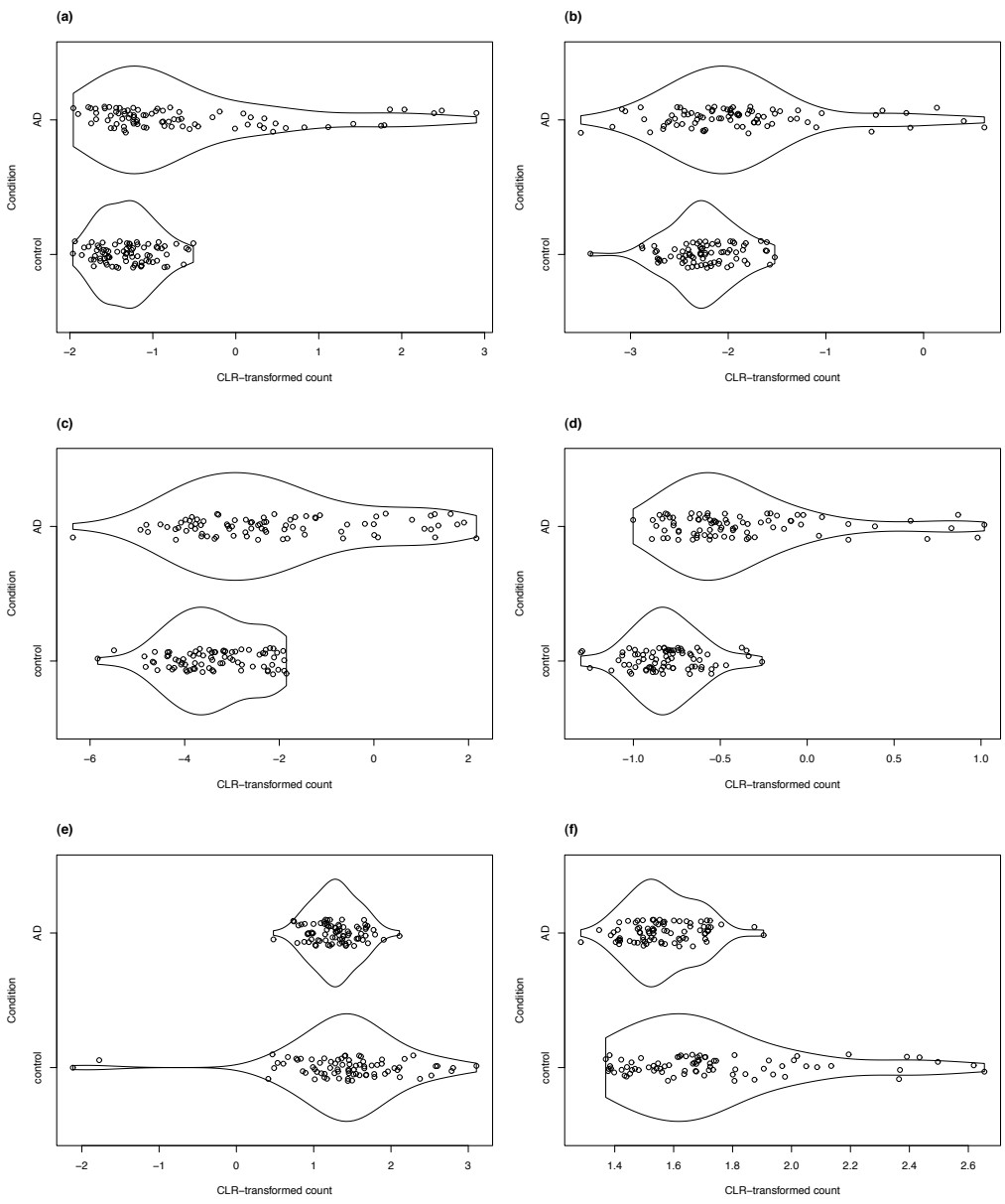

**Figure 8** **Violin plots of selected DV genes detected in the control *vs.* AD comparison.** (A) SLC47A1, (B) C2orf40, (C) SLPI, (D) LTBP2 have the largest SD ratio (> 2); (E) GP1BB and (F) PELP1 have SD ratio about 0.4.

*& Walker, 2019*). *Podvin et al. (2016)* found that C2orf40 is a neuroimmune factor in Alzheimer's disease. The SLC47A1 (solute carrier family 47 member 1) protein is expressed in both the kidney and the brain, and recent research has suggested a linkage between kidney diseases and Alzheimer's disease (*Shi et al., 2018*; *Kelly & Rothwell, 2022*).

We detected 74 genes from the SLC family in the high confidence DV gene set, including four members of the SLC39 family. *Lang et al. (2012)* demonstrated the modulating effect

of dZip1, the ortholog of human SLC39 family transporter, on zinc ion uptake using a *Drosophila* model. Zinc is known to induce amyloid beta formation (*Bush et al., 1994*). Inhibition of dZip1 produces substantial reduction of amyloid beta peptide 42 (A$\beta$42) fibril deposits and less neurodegeneration in A$\beta$42-transgenic flies.

Two of the DV genes with estimated SD ratio substantially smaller than 1 are PELP1 and GP1BB (Fig. 8). PELP1 mediates E2 inhibition of GSK3$\beta$, a neurodegenerative kinase signaling pathway in the brain (*Thakkar et al., 2018*). GSK3$\beta$ is implicated in Alzheimer's disease as a key mediator of cell death (*Llorens-Martin et al., 2014*). The GP1BB gene produces glycoprotein 1b-beta(GPIb$\beta$), a subunit of the GPIb-IX-V protein complex on the surface of platelet cells. Amyloid beta peptides are known to be actively released by platelets (*Bush et al., 1990*; *Casoli et al., 2007*). *Visconte et al. (2020)* recently reported that recruitment of GPIb-IX-V is required for fibrillar amyloid A$\beta$40 and A$\beta$42 to induce platelet aggregation. The study of the role of platelets and the pathogenesis of Alzheimer's disease is an active topic (*Catricala, Torti & Ricevuti, 2012*).

We note that approximately half of genes in the high confidence gene set from the control *vs.* AD comparison (4,271 genes) are also found in the high confidence gene sets from the control *vs.* PSP comparison (4,186 genes). Altogether, 2,149 DV genes are common to both comparisons. This observation is consistent with recent findings that transcriptomic changes are in AD and PSP relative to control are strongly correlated (*Wang et al., 2022*).

## DISCUSSION

Both the results of the simulations and empirical assessment support the feasibility of adopting a CoDA framework for modeling RNA-Seq data. The modeling of CLR-transformed count data using the skew-normal model is justified by the model's good fit. This means that expression variation can be explicitly associated with the standard deviation parameter of the skew-normal model, rather than through a proxy variable such as the dispersion parameter in the negative binomial or the GAMLSS models.

From a practical perspective, the skew-normal model is an ideal choice since it is mathematically tractable and has mature computational support through the sn R package, which enables direct testing of differential variability using its standard deviation parameter. Moreover, a test of differential expression that is based on the mean parameter can also be derived. With these tests, it becomes possible to develop methods for detecting three classes of genes in two-population comparisons: (i) equal variance, different mean; (ii) equal mean, different variance; (iii) different mean, different variance. Although clrDV cannot differentiate genes of the second and the third type, inspection of violin plots should be useful for ascertaining whether the DV genes also appear to differ significantly in the mean of their relative expression level.

We observed that in the comparisons between control *vs.* AD and control *vs.* PSP, a majority of the DV genes identified by clrDV (between 86.1% and 92.0%) were already included in the high-confidence gene set, where the estimated log2(SD ratio) has relatively larger magnitude compared to non-DV genes (Figs. 6C and 6D; Figs. 7C and 7D). Thus, it seems that clrDV alone should be able to recover most of the DV genes of interest.

In standard count models, the choice of normalization method is a hyperparameter that needs to be optimised. It is known that incorrect normalization leads to inflated FDR in differential expression analyses (*Evans, Hardin & Stoebel, 2018*), yet the assumptions that justify a normalization method are usually not testable. Since existing normalization methods have been developed for the purpose of finding differentially expressed genes, the assumptions that justify their use could be suboptimal for the purpose of developing differential variability tests. Consequently, the performance of existing count-based approaches for DV test is likely sensitive to the choice of normalization method. However, it is beyond the scope of the present work to optimize the choice of normalization step for these count-based methods.

On the aspect of practical application, we note that the R codes provided by *De Jong, Moshkin & Guryev (2019)* for GAMLSS are not sufficiently generic and require further user modifications to be suitable for routine use as a DV test. In addition, GAMLSS uses BH rather than BY as the default setting for multiple comparisons adjustment.

For MDSeq, we found that it may occasionally encounter difficulties in estimating model parameters. Specifically, in our analysis of the Mayo RNA-Seq dataset, we observed that 45 genes returned NA parameter estimates in the control *vs.* AD and the control *vs.* PSP comparisons. On the other hand, the Bayesian method DiffDist does not appear attractive for routine use, owing to its long run time and poor control of FDR at larger sample sizes. Therefore, we suggest using GAMLSS-BH for sample sizes of 50 or less, since the simulation results indicate that it has the best statistical power among the methods considered for detecting DV, though with FDR that is likely to be at least 0.05. For larger sample sizes, clrDV or GAMLSS-BY are better alternatives. In fact, users may consider using them jointly to obtain high confidence DV gene sets, as shown in the analysis of the Mayo RNA-Seq dataset.

Finally, the CoDA framework enables the estimation of mean, variance, and skewness parameters simultaneously from RNA-seq data. This unique capability leads to the development of statistical tests that can simultaneously assess differential expression, differential variability, and differential skewness, which is not possible using conventional count modeling techniques. The results of this extension will be reported elsewhere in a separate publication.

## CONCLUSIONS

Variability of gene expression at aberrant levels is one of the hallmarks of disrupted or dysregulated biological processes. Hence, detection of genes with differential variability should accompany routine differential expression analysis to expand the pool of potential therapeutic intervention targets. clrDV offers a novel approach for identifying DV genes in RNA-seq data. By modeling the null distribution of centered log-ratio transformed RNA-Seq data using a skew-normal distribution, clrDV can detect genes with expression variance that differs significantly between two populations. Simulation results demonstrate that clrDV has a comparable or superior false discovery rate and probability of Type II error compared to its close competitors, while also having a faster run time for larger sample sizes

per group. Our analysis of the Mayo RNA-seq dataset revealed that four genes associated with AD pathogenesis (LTBP2, SLPI, C2orf40, and SLC47A) showed significantly higher variance in the AD group. On the other hand, GP1BB and PELP1, which are also linked to AD pathogenesis, showed significantly lower variance in the AD group. In summary, for large RNA-Seq studies (*e.g.*, sample sizes of 100 or more), clrDV's good control of FDR and probability of Type II error, and its ease of implementation *via* the `clrDV` R package make it a good choice for complementing DE tests with DV tests.

## ACKNOWLEDGEMENTS

We wish to thank Dr. C.Y. Ung for helpful discussions. We are grateful to Professor A. Azzalini for developing the theoretical foundations as well as computational tools for the skew normal distribution over many years, without which the present work would not be possible. The results published here are in whole or in part based on data obtained from the AD Knowledge Portal. The Mayo RNAseq study data was led by Dr. Nilüfer Ertekin-Taner, Mayo Clinic, Jacksonville, FL as part of the multi-PI U01 AG046139 (MPIs Golde, Ertekin-Taner, Younkin, Price). Samples were provided from the following sources: The Mayo Clinic Brain Bank. Study data includes samples collected through the Sun Health Research Institute Brain and Body Donation Program of Sun City, Arizona. Finally, we thank the reviewers (Dr. Laura Jenniches and an anonymous reviewer) for their meticulous and constructive comments which helped improve the present article.

### Funding

For the Mayo RNA-Seq dataset, data collection was supported through funding by NIA grants P50 AG016574, R01 AG032990, U01 AG046139, R01 AG018023, U01 AG006576, U01 AG006786, R01 AG025711, R01 AG017216, R01 AG003949, NINDS grant R01 NS080820, CurePSP Foundation, and support from Mayo Foundation. The Brain and Body Donation Program is supported by the National Institute of Neurological Disorders and Stroke (U24 NS072026 National Brain and Tissue Resource for Parkinsons Disease and Related Disorders), the National Institute on Aging (P30 AG19610 Arizona Alzheimers Disease Core Center), the Arizona Department of Health Services (contract 211002, Arizona Alzheimers Research Center), the Arizona Biomedical Research Commission (contracts 4001, 0011, 05-901 and 1001 to the Arizona Parkinson's Disease Consortium) and the Michael J. Fox Foundation for Parkinsons Research. The funders had no role in study design, data collection and analysis, decision to publish, or preparation of the manuscript.

### Grant Disclosures

The following grant information was disclosed by the authors:
NIA grants: P50 AG016574, R01 AG032990, U01 AG046139, R01 AG018023, U01 AG006576, U01 AG006786, R01 AG025711, R01 AG017216, R01 AG003949.
NINDS grant: R01 NS080820.

CurePSP Foundation.
Mayo Foundation.
The National Institute of Neurological Disorders and Stroke.
(U24 NS072026 National Brain and Tissue Resource for Parkinsons Disease and Related Disorders).
The National Institute on Aging (P30 AG19610 Arizona Alzheimers Disease Core Center).
The Arizona Department of Health Services (contract 211002, Arizona Alzheimers Research Center).
The Arizona Biomedical Research Commission (contracts 4001, 0011, 05-901 and 1001 to the Arizona Parkinson's Disease Consortium).
The Michael J. Fox Foundation for Parkinsons Research.

## Competing Interests

The authors declare there are no competing interests.

## Author Contributions

- Hongxiang Li conceived and designed the experiments, performed the experiments, analyzed the data, prepared figures and/or tables, authored or reviewed drafts of the article, and approved the final draft.
- Tsung Fei Khang conceived and designed the experiments, authored or reviewed drafts of the article, and approved the final draft.

## Data Availability

The code is available at GitHub and Zenodo:

-https://github.com/Divo-Lee/clrDV.

- Hongxiang Li. (2023). Divo-Lee/clrDV: Initial Release (0.1.0). Zenodo. https://doi.org/10.5281/zenodo.7900449.

The raw data are available at the AD Knowledge Portal: https://adknowledgeportal.synapse.org/. Permission to use this dataset for publication was granted by the AD Knowledge Database. Controlled Access data (individual-level Human data) is highly sensitive and cannot be redistributed.

## Supplemental Information

Supplemental information for this article can be found online at http://dx.doi.org/10.7717/peerj.16126#supplemental-information.

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
