# Peer review of "clrDV: a differential variability test for RNA-Seq data based on the skew-normal distribution"

_PeerJ, doi:10.7717/peerj.16126_

## Round 0.1 · original submission · Major Revisions

Reviewers suggested major revisions for the manuscript. Please make sure to address all the points and concerns.

·

Basic reporting

The authors developed a tool to compare differential variability of gene expression between two populations. As the authors indicate in their introduction, most of the RNA-seq analysis tools focus on differential expression analysis. Since differential variability has been found to play an important role e.g. in the identification of disease states, a computational tool for this purpose would be desirable. The authors assume that RNA-seq count data is compositional and as such can be described by a skew normal distribution. They published their analysis in an R package which can be accessed on github. Overall, their method is interesting, but the manuscript lacks structure and is very difficult to follow. The study should be entirely rewritten before it could be considered for publication.

With regard to structure, the major problem is the division of the content between Introduction, Methods and Results. Without any apparent reason, the introduction was split into two parts, the second one being called Motivation. This section, however, does not motivate the importance of testing for differential variability, as one would expect, but motivates the choice of compositional data analysis. The formula given there should clearly be moved to Methods. I do not see the need for the separate section Motivation. The content should be shortened and added to the introduction.

Another important issue with regard to structure is the division of content between Results and Methods. The analyses performed in the Results section are generally not motivated. For example at the beginning of the Results (Section 3.1), the authors claim that skew-normal distribution describes CLR-transformed count data well. They base this only on two examples shown in Figure 1, which is insufficient. A reference for this claim or some kind of test involving all genes (i.e. calculating a genome-wide diversion of the data from the distributions and comparing this to a standard normal distribution) would help to make this point. Furthermore, the sample size is varied without explaining what the sample size is. One has to go through the Methods in detail to understand this.

In the Methods, the first section (2.1) is far too long and unstructured. I don’t think the discussion of centered and direct parameters needs to be discussed in this work. It should be sufficient to mention that direct maximum likelihood estimation is not possible for direct parameters such that centered parameters are chosen. In line 139, the authors claim that they show that the skew normal distribution is suited to describe CLR-transformed count data in the supplementary text (S1), which they do not. Instead, they motivate there why centered parameters are used.

The R package clrDV developed and referenced in the manuscript can be downloaded from github. The required data to repeat the analysis can partly be obtained from NCBI Gene Omnibus Expression database. The second dataset used in this study (Mayo RNASeq) can only be used with permission from the AD Knowledge Portal, which the authors have obtained. The R package would profit from an example analysis with a small test dataset provided together with the package. The functions and usage of the R package have not been documented on github.

Language is in general clear enough to be understandable, the readability is rather hindered by the lack of structure, also within a paragraph. I did not correct language at this point because I think that the manuscript has to be entirely restructured and rewritten before being considered for publication.

Figures are generally well chosen, but font size should be increased. Figure 2 should show fewer examples, two different sample sizes should be sufficient. Or maybe the authors could plot mean Type II Error and Mean FDR for every tool depending on sample size into one figure instead. The same applies to Figure 3. For Figure 4, I would use histograms rather than volcano plots. That would make the message clearer.

Experimental design

The authors did not perform any experiments themselves. The choice of the analyzed datasets was not motivated. For example in Section 3.1, the authors say that they are using the skew normal distribution and refer to Figure 1 before they even introduce the dataset that they work with. The same is true for the Mayo dataset in Section 3.2. Therefore, even though the overall research question (development of an R package for differential variability analysis of count data) is clear, the intermediate steps that were taken to reach that goal are not well motivated. Since this motivation is missing, I cannot evaluate whether or not the datasets were chosen adequately.
It seems that the authors make their full analysis available along with the R package clrDV, which is great. Their benchmarking of the various tools looks good, even though the choice of sample size and other parameters is not motivated in Section 2.3. I also disagree with the authors that run time alone should be a reason to exclude DiffDist from the analysis. The authors could have used it on a subset of their simulations. A very high accuracy for small sample size would have justified long run times.

Validity of the findings

Having a standard tool for differential variability of RNA-seq/count data analysis would be desirable. To become a standard tool, extensive documentation on github along with self-contained demos are required. In addition, the tool only reaches the accuracy of existing tools (DSeq and GAMLSS-BY, Figure 3 and 4) for large sample sizes. Due to experimental restrictions (costs, time) sample sizes might often not reach this size (if the authors disagree with that, they need to motivate the chosen sample sizes properly). For smaller sample sizes, MDseq and GAMLSS-BY have faster run time and higher accuracy. A rigorous analysis of when clrDV should be used should be performed to advise potential users when to use this package.

Additional comments

I will include an annotated PDF file with some comments. I did not comment on the whole text because the short-comings indicated above are so major that the manuscript needs to be entirely rewritten. Nevertheless, the availability of such a tool would be great, therefore the publication of this work should be considered after major revisions.

Reviewer 2 ·

Basic reporting

The authors have written the article with professional english and did a thorough background research. Differential variability (DV) genes represent genes that exhibit significant variability differences between disease and control groups, even if their mean expression levels remain unchanged. These genes have been shown to play important roles in various biological processes and to be associated with several diseases. However, one point is not quite clear for me. Which is the biological implications behind DV genes. The authors have given examples that showing DV genes can explain certain biological phenomenon. For example in DE genes, they are affected by transcription factors that are dysregulated in certain diseases and consequently affect biological pathways. Please address in DV genes, what could be the possible biological events that causes DV genes? And why is it relevant to the diseases?
Minor comment:
- Can you please describe the run time in terms of complexity? (line 235 - 236)
- In line 244, (After filtering, sample sizes of the control...), remove the comma and state again the what is the filtering criteria. Therefore, the revised sentence could be: "After filtering out samples with (the criteria), the sample sizes of the groups were ....."

Experimental design

Methods section is very well-stated and I appreciate the immense work has been put in it. Please add a comment of why the authors have chosen Benjamini-Yekutieli method for mutliple-testing correction? Would you consider providing other methods?
Minor comment: Please provide a thorough vignettes for usage guidance for your package.

Validity of the findings

The authors have shown that clrDV package is better in detecting DV genes considering with lower false discovery rate. Major comment: Please consider performing a DE analysis and GO terms enrichment analysis to address the qualitative difference of using DV and DE analysis.

---

## Round 0.2 · Major Revisions

Please address the comments from the reviewer especially correcting the structure of the manuscript. Please make sure that the methods section should only contain short and precise descriptions of the methods and the motivation, and background for these methods should be given in the introduction and results section.

·

Basic reporting

The authors have reacted to many of the comments, but especially with regard to structure, some more work is necessary. The methods section should contain a short and precise description of the methods, but the motivation for the methods should be in the introduction and results section. This affects all methods sections. For example in line 122, the authors start the methods section with a historical description of where compositional data analysis comes from. That should rather be done in the introduction.

Since the authors motivate their methods in the methods section, the results section is too short. It would be helpful if the authors started the results section with introducing and motivating their methods and describing their workflow (for formulas, they should refer to the method section). In line 270, the authors interpret Figure 1 without describing first what it shows. They could start by relating why they use compositional data and the skewed normal distribution. Then, they could explain why they compare the skewed normal distribution to the distribution of read counts.

Related to these structural issues, the style could be improved. The results section is mostly a description of the figures without motivating the procedures. For example in line 273, the authors do start by describing Figure 2 instead of motivating why they perform this comparison and what their goals are.

With regard to my previous comment on Figure 2-4, I would suggest the authors create summary figures and move the scatter plots to the supplementary material. Then, they can also present the results for n=125 and n=250.

The authors are right that changing Figure 5 to histograms would remove the information on how biologically relevant the results are.

Experimental design

The authors have reacted to my comments on the first version of the manuscript. The design of the analysis seems fine if the authors comment on the point that I make in "Validity of the findings".

Validity of the findings

The authors have reacted to my comments on the first version of the manuscript. The fact that there is a strong correlation between effect size and adjusted p value in Figure 5 suggests some systematic problem with the procedure. The authors should look into this again and at least comment on it in the text if they believe that this is not the case.

---

## Round 0.3 · accepted · Accept

Your revised manuscript was found to be acceptable for publication. Please follow any other requests and reviews from our journal editorial office.
Sincerely.

Reviewer 3 ·

Basic reporting

The manuscript has been improved in terms of structure and clarity. In the previous review, there were concerns about the placement of methodological motivations, particularly the historical description of where compositional data analysis originates. The authors have now addressed this by relocating "Section 2.1 The compositional data analysis framework" from the methods section to the introduction. This change enhances the flow and coherence of the manuscript, ensuring that readers are introduced to the foundational concepts before delving into the specific methods.

Experimental design

The experimental design of the study is robust. The authors have provided a clear rationale for their methodologies, emphasizing the significance of the skew-normal model in capturing variation in CLR-transformed counts of genes. In the revised manuscript, they have made efforts to highlight the performance of clrDV in comparison to existing methods for detecting differential variability. This comparative approach strengthens the credibility of their experimental design and findings.

Validity of the findings

The findings presented in the manuscript are supported by rigorous analysis. The authors have showcased the practical utility of clrDV, demonstrating its capability to identify genes associated with specific biological conditions, such as Alzheimer’s disease. In the previous review, there was an emphasis on the value of the skew-normal model in relation to model parameters. The authors have now further elucidated this by associating expression variation explicitly with the standard deviation parameter of the skew-normal model.

Additional comments

The authors have been proactive in addressing feedback from the previous review. They have made several changes to improve the manuscript's flow and clarity.